 

🔓 | **Open Peer Review** | Clinical Microbiology | Research Article

# Genome sequence-based species classification of *Enterobacter cloacae* complex: a study among clinical isolates

Xuemei Qiu,[1,2] Kun Ye,[1] Yanning Ma,[1] Qiang Zhao,[1] Lifeng Wang,[1] Jiyong Yang[1]

**ABSTRACT** Accurate species-level identification of *Enterobacter cloacae* complex (ECC) is crucial for related research. The classification of ECC is based on strain-to-strain phylogenetic congruence, as well as genomic features including average nucleotide identity (ANI) and digitalized DNA-DNA hybridization (dDDH). ANI and dDDH derived from whole-genome sequencing have emerged as a reliable metric for assessing genetic relatedness between genomes and are increasingly recognized as a standard for species delimitation. Up to now, there are two different classification methods for ECC. The first one categorizes *E. hormaechei*, a species within ECC, into five subspecies (*E. hormaechei* subsp. *steigerwaltii,* subsp. *oharae,* subsp. *xiangfangensis,* subsp. *hoffmannii,* and subsp. *hormaechei*). The second classifies *E. hormaechei* as three species: *E. hormaechei, "E. xiangfangensis," "E. hoffmanii."* While the former is well-accepted in the academic area, the latter may have a greater ability to distinguish different species of ECC. To assess the suitability of these identification criteria for clinical ECC isolates, we conducted a comprehensive analysis involving phylogenetic analysis, ANI and dDDH value alignment, virulence gene identification, and capsule typing on 256 clinical ECC strains isolated from the bloodstream. Our findings indicated that the method of categorizing *E. hormaechei* into five subspecies has better correlation and consistency with the molecular characteristics of clinical ECC isolates, as evidenced by phylogenetic analysis, virulence genes, and capsule typing. Therefore, the subspecies-based classification method appears more suitable for taxonomic assignments of clinical ECC isolates.

**IMPORTANCE** Standardizing taxonomy of the *Enterobacter cloacae* complex (ECC) is necessary for data integration across diverse studies. The study utilized whole-genome data to accurately identify 256 clinical ECC isolated from bloodstream infections using average nucleotide identity (ANI), digitalized DNA-DNA hybridization (dDDH), and phylogenetic analysis. Through comprehensive assessments including phylogenetic analysis, ANI and dDDH comparisons, virulence gene, and capsule typing of the 256 clinical isolates, it was concluded that the classification method based on subspecies exhibited better correlation and consistency with the molecular characteristics of clinical ECC isolates. In summary, this research contributes to the precise identification of clinical ECC at the species level and expands our understanding of ECC.

**KEYWORDS** *Enterobacter cloacae* complex, average nucleotide identity, DNA-DNA hybridization, whole-genome sequencing

*E*nterobacter cloacae complex (ECC) has emerged as a primary cause of opportunistic infections, especially in immunocompromised individuals (1). Given the diverse nature of resistance, virulence, and disseminating among different ECC species and subspecies (2, 3), precise assignation of clinical isolates at species and subspecies levels is essential for a comprehensive understanding of the epidemiology, pathogenesis, and microbiological features of these bacteria.

Address correspondence to Jiyong Yang, yangjy301@163.com.

The authors declare no conflict of interest.

See the funding table on p. 8.

The taxonomy of ECC has evolved over time, with bacterial species assignment tools transitioning from phenotypic methods (4), such as Gram staining, biochemical assays, and mass spectrometry (MS), to molecular approaches, such as DNA-DNA hybridization (DDH) (5) and 16S rRNA gene sequencing (6) and marker gene *hsp60* sequencing (7). While MS has been commonly used in clinical laboratories, it has been proven inaccurate for ECC classification (8, 9). Thus, low-cost whole-genome sequencing (WGS)-based taxonomic approaches, specifically average nucleotide identity (ANI) and digitalized DNA-DNA hybridization (dDDH), have gained widespread use for precise species identification (10). ANI and dDDH derived from whole-genome sequencing have emerged as a reliable metric for assessing genetic relatedness between genomes and are increasingly recognized as a standard for species delimitation. The recommended ANI threshold to define species was adjusted to a range of around 95%–96% (10, 11). However, there are currently two classifications: one based on subspecies, which categorizes *E. hormaechei* into five subspecies and the other divides *E. hormaechei* into three species.

One method classifies *E. hormaechei* into five subspecies (*steigerwaltii*, *oharae*, *xiangfangensis*, *hoffmannii*, and *hormaechei*). Hoffmann and Roggenkamp utilized marker genes, particularly *hsp60*, to identify clusters within the ECC (7). Subsequently, Hoffman et al. investigated three closely grouped clusters and delineated the current three subspecies (*steigerwaltii*, *oharae*, *hormaechei*) of *E. hormaechei* based on DDH and phenotypic tests (12). Chavda et al. employed ANI and single nucleotide polymorphisms (SNPs) from whole-genome alignments to establish 18 groups (groups A to R) within ECC (13). They found that the mean ANI values between 18 ECC groups were always ≤95%, except among *E. hormaechei* subspecies groups A to E, G, and H (13). Sutton et al. performed a more detailed ANI analysis across all *Enterobacter* genomes in NCBI's RefSeq, supporting Chavda's groups A–E as subspecies of *E. hormaechei*. The five subspecies (*steigerwaltii*, *oharae*, *xiangfangensis*, *hoffmannii*, and *hormaechei*) of *E. hormaechei* defined by Sutton et al. fell within the expected ANI range for bacterial species, with ANI > 95% between subspecies and >98% ANI with a subspecies (14). Thus, this classification adheres to ANI > 95% for species and ANI > 98% for subspecies, as well as dDDH value >70% with the type strains.

Recently, researchers proposed a novel perspective (Species: ANI > 96% and dDDH >70%), classifying *E. hormaechei* into three species: *E. hormaechei*, "*E. xiangfangensis*," and "*E. hoffmanii*" (15, 16). Ciufo et al. used a fast *k*-mer comparison based on MinHash to compare prokaryotic genomes at the NCBI with existing assemblies. They found that ANI > 96% can discriminate the most strains (prokaryotic genomes in the NCBI) into distinct species, and it is observed that a lower ANI value correlates with an increased likelihood of ambiguous classification outcomes (17). Zong et al. used cutoff of ANI ≥ 96% and dDDH ≥ 70.0% for the type strain of *Enterobacter* (*E. hormaechei* subsp. *oharae* DSM 16687, *E. hormaechei* subsp. *steigerwaltii* DSM 16691, "*E. xiangfangensis*" LMG 27195). The ANI of three type strains was ≥96.62%, and the dDDH was ≥75.8%. Consequently, they classified the three type strains as one species (15). They found that all *Enterobacter* subspecies assignments were incorrect and updated the taxonomy of *Enterobacter*. *E. hormaechei* were classified into three species ("*E. xiangfangensis*," "*E. hoffmanii*," and *E. hormaechei*).

The existence of these two perspectives to categorize ECC has introduced confusion in practice. Accurate classification of ECC species is essential for conducting *in vitro* and *in vivo* phenotype and function verification experiments and providing detailed and reliable data for subsequent clinical applications (3). Therefore, we analyzed 256 clinical ECC isolates from the bloodstream using ANI and dDDH, constructed a phylogenetic tree using kSNP, and then conducted a comprehensive analysis to assess the suitability of these classification criteria for clinical ECC isolates.

## MATERIALS AND METHODS

### Clinical isolates

We collected 256 nonduplicated clinical strains from patients with ECC-positive blood cultures at the First Medical Center of Chinese PLA General Hospital from 2010 to 2022. The isolates were stored in 30% glycerol at −80°C until further analysis. Initial species identification was performed using VITEK MALDI-TOF MS (bioMérieux). The isolates were identified as *E. asburiae*, *E. hormaechei*, *E. kobei*, *E. ludwigii,* or *E. cloacae*. ECC isolates were routinely on Columbia blood agar with aeration at 37°C. Bloodstream-isolated ECC were distributed at all ages, and the largest number of strains were middle-aged (41–65 years old) and elderly (>66 years old), with 116 strains (45.31%) and 94 strains (38.67%), respectively.

### Sequencing, assembly, and annotation of bacterial genomes

All isolates were subjected to WGS using a paired-end library with an average insert size of 350 bp (ranging from 150 to 600 bp) on a HiSeq sequencer (Illumina, CA, USA). Sequence quality was evaluated using fastqc and fastp, the N-base, $Q$-value < 30 and poor quality reads were removed. Sequence was assembled with SPAdes (SPAdes-3.15.5). The quality after assembly was evaluated and processed with Trimmotic and seqkit; contigs shorter than 200 bps were removed from the results. Multi-locus sequence typing (MLST) analysis was conducted using mlst v.2.19.0 (https://github.com/tseemann/mlst) and PubMLST (18). Virulence genes were identified using VFDB databases (19), the coverage ≥80% and identity ≥90%. Capsule typing was carried out with fastKaptive v0.7.3 (20) using an *Enterobacteriaceae* cps database (21).

### ANI and digital DNA:DNA hybridization analysis

The ANI was analyzed with fastANI (22), and the type strains of *Enterobacter* species and subspecies were collected from NCBI Refseq Database. The type strains (Table S1) were used for the ANI analysis. The dDDH among 256 clinical isolates was determined using the web-service https://tygs.dsmz.de/ (23).

### Phylogenetic analysis

The phylogenetic analysis was performed using kSNP3, a *k*-mer-based tool. Using Kchooser, an optimal *k*-mer size of 19 was determined. The genomic distances were calculated using core SNPs to construct the phylogenetic tree. The results were visualized using iTOL editor v5 (24).

## RESULTS

### Species identification of clinical isolates

When using a 95% and 98% ANI cutoff to define species or subspecies boundaries, the 256 clinical strains isolated from bloodstream were classified as *E. hormaechei* (*n* = 162), *E. kobei* (*n* = 20), *E. bugandensis* (*n* = 17), *E. asburiae* (*n* = 14), *E. ludwigii* (*n* = 13), *E. roggenkampii* (*n* = 10), *E. cloacae* (*n* = 9), *E. chengduensis* (*n* = 3), *E. mori* (*n* = 3), *E. quasihormaechei* (*n* = 3), *E. chuandaensis* (*n* = 1), *E. sichuanensis* (*n* = 1), and *E. wuhouensis* (*n* = 1). Moreover, the 162 *E. hormaechei* isolates were further classified into five subspecies: *E. hormaechei* subsp. *steigerwaltii* (*n* = 71), *E. hormaechei* subsp. *xiangfangensis* (*n* = 50), *E. hormaechei* subsp. *hoffmanii* (*n* = 33), *E. hormaechei* subsp. *oharae* (*n* = 6), and *E. hormaechei* subsp. *hormaechei* (*n* = 1). The most common species in clinical samples was *E. hormaechei* (Fig. 1).

When using a 96% ANI cutoff to define species boundaries, the 256 clinical strains were classified as same as described above plus *"E. xiangfangensis"* (*n* = 127), *"E. hoffmanii"* (*n* = 33), and *E. hormaechei* (*n* = 1). In other words, the 162 isolates initially classified as *E. hormaechei* under the 95% ANI cutoff were reclassified into three species

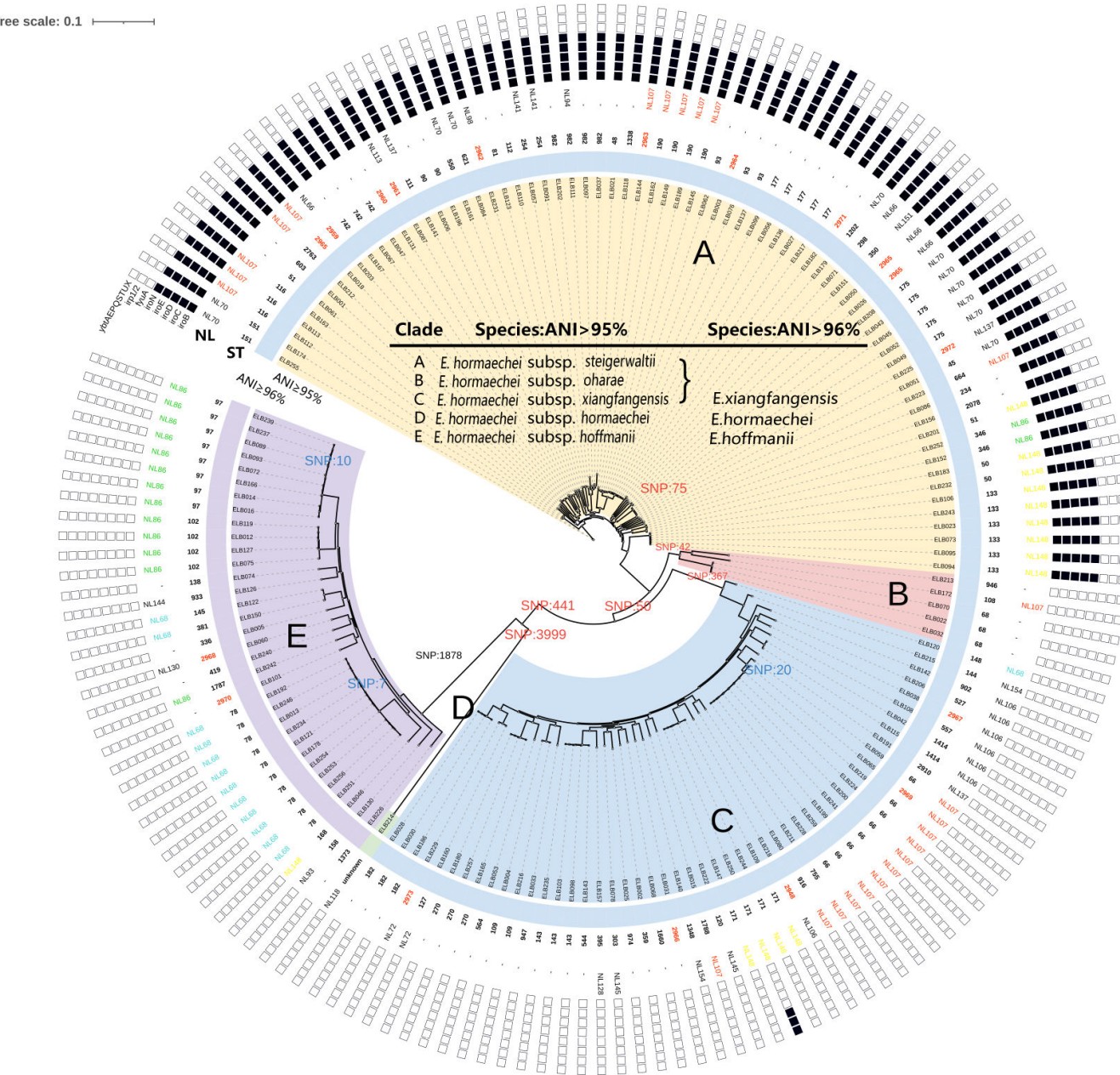

**FIG 1** Species identification, STs, capsule typing, and virulence genes detection of *E. hormaechei*. The species/subspecies and clades are drawn in concentric circles. ST, multi-locus sequence typing. New STs identified in this study are indicated in red font. NL, capsule typing of *Enterobacter*. Black and white squares represent the presence and absence of siderophores-related genes, respectively. The genomic distance has shown based on SNPs.

("*E. xiangfangensis*," "*E. hoffmanii*," and *E. hormaechei*). The most common species in clinical samples appears to be "*E. xiangfangensis*" (Fig. 1). Therefore, we conducted a detailed analysis of the 162 strains with dispute classification.

For the ECC strains in our study, we also used dDDH for species classification, and the results were consistent with the results of species and subspecies classification using a 95% and 98% ANI cutoff to define species or subspecies boundaries.

## Phylogenetic analysis and sequence types

Phylogenetic analysis using kSNP3, a *k*-mer-based tool, showed that the 162 clinical strains isolated from the bloodstream were clustered into five clades (Fig. 1A through E).

Each of the five clades corresponded to one of the five subspecies classified based on the 95% ANI cutoff: *E. hormaechei* subsp. *steigerwaltii,* subsp. *oharae,* subsp. *xiangfangensis,* subsp. *hoffmannii,* and subsp. *hormaechei*, respectively (Fig. 1). The genomic distance based on SNP between the five clades and within each clade has shown in the figure (Fig. 1). Remarkably, the 162 isolates of *E. hormaechei* were assigned to 70 different STs, with the most prevalent being ST78 (3.9%, 10/257), ST66 (3.1%, 8/257), and ST97 (3.1%, 8/257).

## Virulence genes analysis

Virulence genes (encoding fimbriae, curli fibers, capsules, and flagella) were widely distributed in 162 strains with no significant differences. Additionally, siderophore-related genes [*entABESfepABCDG* (enterobactin) and *iutAiucABCD* (aerobactin)] were present in all 162 clinical isolates. However, only 72 strains within the yellow clade (clade A) carried the *iroBCDEN* (salmochelin) gene cluster. This clade aligned with *E. hormaechei* subsp. *steigerwaltii*, as identified using the 98% ANI cutoff to define subspecies boundaries (Fig. 1). Surprisingly, our analysis, for the first time, revealed the exclusive occurrence of *iroBCDEN* gene clusters all in *E. hormaechei* subsp. *steigerwaltii* when strains were identified as *E. hormaechei*. However, when using the 96% ANI cutoff to delineate species in the ECC, strains in three different color clades were grouped under the specie *"E. xiangfangensis,"* each harboring chromosomes with several virulence gene clusters encoding different types of siderophores (Fig. 1).

Furthermore, we analyzed 6,221 genomes of *E. hormaechei* from the NCBI database (update to September 2023). Our results revealed that *iroBCDEN* cluster was detected only in 2,722 genomes of *E. hormaechei* subsp. *steigerwaltii* defined by the 95% ANI cutoff, but not in the other four subspecies. This finding validates the earlier statement that the *iroN* gene was specific to *E. hormaechei* subsp. *steigerwaltii* when strains were identified as *E. hormaechei*.

Meanwhile, we looked at the clinical outcomes of patients with bacteremia in different subspecies of *E. hormaechei* and found no significant differences in disease severity and mortality, regardless of whether this *iroBCDEN* virulence gene cluster was present.

## Capsule typing

Among all ECC strains, 66.41% (170/256) were classified into 48 capsule typing. When using the 95% ANI cutoff to define species boundaries, 108 isolates of *E. hormaechei* (108/162, 66.67%) contained complete cps locus. (Fig. 1). Of note, NL148 (9/14) and NL70 (11/11) were predominantly distributed in *E. hormaechei* subsp. *steigerwaltii*, NL106 (7/7) in *E. hormaechei* subsp. *xiangfangensis*, while NL86 (13/15) and NL68 (12/13) were prevalent in *E. hormaechei* subsp. *hoffmannii*. Overall, different *E. hormaechei* subspecies exhibited different dominant serotypes (Fig. 1). When using the 96% ANI cutoff to define species boundaries, 29 isolates of *"E. hoffmanii"* (*n* = 33) were classified into *Enterobacter*-NL86 (*n* = 15) and *Enterobacter*-NL68 (*n* = 13) (Fig. 1).

## ANI and dDDH

We performed pairwise, whole-genome comparisons among 256 sequenced clinical ECC isolates to calculate ANI values. According to the established standard, where ANI > 95% and >98% were considered accurate thresholds for demarcating species and subspecies, respectively, two clinical strains (ELB120 and ELB211) were classified as *Enterobacter hormaechei* subsp. *xiangfangensis*. We used dDDH for species classification; ELB120 and ELB211 were consistent with the results using a 95% and 98% ANI cutoff to define species or subspecies and classified as *Enterobacter hormaechei* subsp. *xiangfangensis*. However, when using ANI > 96% as the standard for identifying ECC species, clinical strains (ELB120 and ELB211) were classified as *"E. xiangfangensis"* or *"E. hoffmanii"* (Table 1), thereby introducing confusion into the classification.

**TABLE 1** Identification of two clinical strains with different ANI values[a]

| Isolate | Species: ANI > 95%, subspecies: ANI > 98% | Species: ANI > 96% | ANI |
|---------|---------------------------------------------|--------------------|-----|
| ELB120 | *E. hormaechei* subsp. *xiangfangensis* | *E. xiangfangensis* | **98.683929** |
| | *E. hormaechei* subsp. *oharae* | | 97.407921 |
| | *E. hormaechei* subsp. *steigerwaltii* | | 97.342178 |
| | *E. hormaechei* subsp. *hoffmannii* | *E. hoffmanii* | **96.015724** |
| | *E. hormaechei* subsp. *hormaechei* | *E. hormaechei* | 95.021317 |
| ELB211 | *E. hormaechei* subsp. *xiangfangensis* | *E. xiangfangensis* | **99.152214** |
| | *E. hormaechei* subsp. *oharae* | | 97.106842 |
| | *E. hormaechei subsp. steigerwaltii* | | 97.041702 |
| | *E. hormaechei subsp. hoffmannii* | *E. hoffmanii* | **96.005402** |

[a]The bold ANI values mean that clinical strains (ELB120 and ELB211) can be identified as "E. xiangfangensis" and "E. hoffmanii" using the 96% ANI cutoff to define species boundaries.

## DISCUSSION

Accurate subspecies and species identification of important pathogens is pivotal for these pathogens differ in their biological and pathological characteristics (8, 25). With the tools of species classification continually improving (26) and a significant reduction in costs, ANI and dDDH emerged as widely utilized and recommended methods for determining species boundaries. Currently, the taxonomic assignment of ECC encompasses two distinct perspectives, both coexisting in recent literature publications (27, 28). The choice of classification method is crucial for subsequent epidemiological, molecular, and pathogenicity analyses. Our preliminary species classification revealed a focal point of controversy in naming 162 clinical strains isolated from the bloodstream, prompting a detailed analysis to discern the potential clinical implications of these two taxonomic assignments.

kSNP is able to identify core SNPs in the genome sequence and estimate phylogenetic trees based on these SNPs found in all genomes. It is suitable for our study to classify *E. hormaechei* based on the high similarity among their genomes (29). Our findings revealed that the 162 clinical *E. hormaechei* strains isolated from the bloodstream were divided into five distinct clades (A-E, Fig. 1), aligning with subspecies-based taxonomy (ANI > 95% for species and ANI > 98% for subspecies). It has been documented that the *k*-mer-based core SNPs approach exhibited a high contrast enhancement effect in analyzing and identifying small differences, thereby providing a valuable perspective for constructing evolutionary trees (29). We did ANI clustering analysis and the results of 162 *E.hormaechei* clinical isolates were consistent with kSNP approach. kSNP phylogenetic analysis supports our classification based on subspecies from another perspective.

Virulence-associated genes identified in 162 *E. hormaechei* strains isolated from the bloodstream exhibited diverse genomic profiles. Multiple virulence genes were identified in the chromosome of *E. hormaechei* clinical isolates, including the siderophores gene clusters *entABES* and *iutAiucABCD,* encoding enterobactin and aerobactin, respectively (30). The bacterial iron acquisition mechanism contributes to virulence and is important for bacterial metabolism and infection. Meanwhile, *entB*, *iroN,* and *ybtS* genes are involved in siderophore production (31, 32). Our study found that *iroBCDEN* gene clusters were only present in the chromosome of 72 *E. hormaechei* subsp. *steigerwaltii* clinical strains. These clusters encode siderophore salmochelin, facilitating the iron acquisition and evasion of host-defense protein lipocalin-2 (33–35). Moreover, certain *E. bugandensis* isolates harbored *iroN* gene clusters, leading to the death of 6 out of 7 neonates with bacteraemia in a neonatal intensive care unit (NICU) over a 13-month period (36). This underscores the clinical significance of distinguishing strains with and without specific virulence gene clusters, such as *iroN* gene clusters, in clinical pathogenicity analyses. However, there were no significant differences in disease severity and mortality among patients with bacteremia of different subspecies of *E. hormaechei*, regardless of the presence or absence of the *iroBCDEN* gene cluster. The pathogenicity of the strains is influenced by many factors and regulatory mechanisms. The presence of salmochelin in

our isolates may not affect the outcomes of patient. The use of 96% ANI threshold based on WGS for demarcating ECC species consolidating *E. hormaechei* subspecies (subsp. *steigerwaltii*, subsp. *oharae*, and subsp. *xiangfangensis*) into a single species *"E. xiangfangensis"* complicated our understanding of the virulence phenotype and transmission ability of *"E. xiangfangensis"* and negatively impacted subsequent clinical pathogenicity analyses of the species.

Polysaccharide capsule is a key immune evasion determinant for *E. hormaechei* (37). It enhances bacterial resistance to phagocytosis, antimicrobial peptides, and complement deposition under *in vitro* conditions (38, 39). Meanwhile, different capsule types play a decisive role in determining the virulence traits and overall virulence level of *K. pneumoniae* (40, 41). We identified the capsule types of the 162 clinical isolates from the bloodstream. Our results revealed inconsistent distribution of the main serotypes. Different *E. hormaechei* subspecies exhibited different dominant capsule types, suggesting that the virulence potential may vary among *E. hormaechei* subspecies. Moreover, we observed a strong association between MLST and capsule type. In all 170 of the 256 sequence- and capsule-typed assemblies, a given ST consistently correlated with a distinct capsule type. These results indicated that various STs could be grouped into a serotype with identical *cps* loci (Fig. 1). This novel perspective provides valuable insights for the comprehensive study of bacterial virulence and pathogenicity.

Routine identification techniques, such as mass spectrometry, failed to accurately identify and distinguish *E. hormaechei* from *E. cloacae* (42). Digital DNA:DNA hybridization was the major criterion used to justify ANI implementations and thresholds for species delineation in accordance with the recommendations by the *ad hoc* committee for the re-evaluation of the species definition in bacteriology (16, 43). For the ECC strains in our study, we also used dDDH for species classification, and the results were consistent with the results of species and subspecies classification using a 95% and 98% ANI cutoff to define species or subspecies. Estimating the genetic relatedness among whole-genome sequences of the various ECC species through ANI has facilitated reevaluation of the genus phylogeny, shedding light on cases where misidentification occurred with routine techniques. The use of ANI >96% as the threshold for species classification introduces confusion, exemplified by clinical strains like isolates ELB120 and ELB211, which were alternatively classified as *"E. xiangfangensis"* or *"E. hoffmanii."* Although choosing the one with a higher ANI value is an option, it inevitably necessitates a decision. At the same time, it is necessary to study other parameters, such as dDDH and phylogenetic analysis of phylogenetic positions in the evolutionary tree.

## Conclusion

We confirmed distinctions in molecular characteristics and virulence potential among various subspecies within a species by analyzing clinical strains. Standardizing species classification is necessary for data integration across diverse studies. The molecular identification method with an ANI cutoff (ANI >95% for species and >98% for subspecies) and dDDH >70% divides *E. hormaechei* into five subspecies. The subspecies-based classification method appears more suitable for taxonomic assignments of clinical ECC isolates.

## ACKNOWLEDGMENTS

This work was supported by grants from National Key R&D Program of China (MOST, 2022YFC2403304).

X.Q. performed molecular genetic studies, analyzed experimental data, and drafted the manuscript. K.Y., Y.M., Q.Z., and L.W. participated in strain isolation and antimicrobial susceptibility testing. J.Y. designed the study and revised the manuscript. All authors have read and approved the final manuscript.

## AUTHOR AFFILIATIONS

[1]Laboratory Medicine Department, First Medical Center of Chinese PLA General Hospital, Beijing, China
[2]Medical school of Chinese PLA, Beijing, China

## AUTHOR ORCIDs

Xuemei Qiu http://orcid.org/0009-0000-9682-0967
Jiyong Yang http://orcid.org/0000-0002-1194-175X

## FUNDING

| Funder | Grant(s) | Author(s) |
| --- | --- | --- |
| MOST \| National Key Research and Development Program of China (NKPs) | 2022YFC2403304 | Jiyong Yang |

## DATA AVAILABILITY

The bacterial genome data haves been deposited in NCBI under BioProject accession PRJNA1046721.

## ADDITIONAL FILES

The following material is available online.

Supplemental Material

Table S1 (Spectrum04312-23-s0001.docx). Type strain genomes for ECC.

Open Peer Review

PEER REVIEW HISTORY (review-history.pdf). An accounting of the reviewer comments and feedback.

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
