## [Reviewer comments · Microbiology Spectrum]

Microbiology Spectrum

Genome Sequence-Based Species Classification of *Enterobacter cloacae* complex: a Study among Clinical Isolates

Jiyong Yang, Xuemei Qiu, Kun Ye, Yanning Ma, Qiang Zhao, and Lifeng Wang

Corresponding Author(s): Jiyong Yang, First Medical Center of Chinese PLA General Hospital

Review Timeline:

Submission Date:	December 28, 2023
Editorial Decision:	February 10, 2024
Revision Received:	February 27, 2024
Editorial Decision:	March 18, 2024
Revision Received:	April 3, 2024
Accepted:	April 13, 2024

Editor: Sadjia Bekal

Reviewer(s): Disclosure of reviewer identity is with reference to reviewer comments included in decision letter(s). The following individuals involved in review of your submission have agreed to reveal their identity: Francois Gravey (Reviewer #2)

Transaction Report:

DOI: <https://doi.org/10.1128/spectrum.04312-23>

Re: Spectrum04312-23 (Genome Sequence-Based Species Classification of *Enterobacter cloacae* complex: a Study among Clinical Isolates)

Dear Dr. Jiyong Yang:

Thank you for the privilege of reviewing your work. Below you will find my comments, instructions from the Spectrum editorial office, and the reviewer comments.

Revision Guidelines

Sincerely,
Sadjia Bekal
Editor
Microbiology Spectrum

Reviewer #1 (Comments for the Author):

The manuscript entitled "Genome Sequence-Based Species Classification of *Enterobacter cloacae* complex: a Study among Clinical Isolates" reports a study, which is genome sequence-driven identification of 200+ clinical strains within the ECC. However, the ANI values were used as the only deciding parameter for species and subspecies level identification of the bacterial strains. The unavailability of sequence accession numbers makes it more difficult to convince with the overall

conclusions of the study. The study included ECC strains collected from the year 2010 to 2022, which is a big positive, but no information on the related metadata, like the origin of strains is missing. The initial identification was done by MALDI-TOF MS but the results were not discussed at all. In addition to these major comments, I also have several specific comments:

L60-62: There are no two different ANI thresholds, it is just one range from 95-96%. As it has been found that different methods and tools of ANI calculations can lead to variations, this range of ANI values is considered a threshold.

Line 62: Several recent studies have suggested that the subspecies delineations are not accurate as there is no evidence of genetic isolation between them. One can take the example of *Klebsiella pneumoniae*, where three of its subspecies are now considered just *Klebsiella pneumoniae*. In the case of ECC, there is plenty of evidence that the five subspecies of *E. hormaechei* are divided into three species of *Enterobacter*, please refer to the latest research in this area.

Line 69: Granger et al. is missing from the list of references.

Line 72-73: Add reference here

Line 73: The species "*E. xiangfangensis*" and "*E. hoffmanii*" are not yet validated, therefore always express them under "".

L76-77: Both ANI and dDDH values indicated the species delineation, which is further substantiated by the recent core-gene phylogeny of ECC (see the recent publications on this).

L78-79: Authors are wrongly interpreting the ANI range, I would like to mention once again this is a range not two different thresholds. This range helps when the different ANI values result from different ANI calculation tools, which follow slightly different methods of ANI calculations.

The whole Material and Methods sections need more details, it is very brief in the present form.

L94-95: There is a need to explain what are quality criteria used for taking the sequencing data further for assembly.

L100-101: I think more information is required on how the dataset has been generated for type strains because in some instances NCBI RefSeq database can also have taxonomic errors. It is advised to define the whole process of dataset creation, which was further used for ANI calculations and phylogenetic analysis.

I do not understand why the authors ignored the dDDH calculations, as it is still very relevant.

L135-137: No information is provided on the sequence similarity cutoff used for virulence gene assignments.

L164-169: Authors are making an overstatement here, such situations are likely to happen, as there will be always a few strains that are almost on the border of a species, and in such cases the high closest similarity with enough dissimilarity with the second closest need to be considered before making conclusions. Secondly, there is a need to look into other parameters, like dDDH and phylogenetic position in core-gene trees or species trees.

Review of: *Genome Sequence-Based Species Classification of Enterobacter cloacae complex: a Study among Clinical Isolates*

Qiu and colleagues compared two taxonomic approaches for *Enterobacter cloacae* complex genomes. These two were based on Average Nucleotide Identity (ANI) but the thresholds applied for the species definition could be: ANI > 95% and subspecies >98% or ANI > 96% for species definition and no data for subspecies.

To test the two thresholds, they used a collection of 256 clinical strains isolated from blood cultures between 2010 and 2022. A global phylogenetic approach was performed using the kSNP. For each genome, several typing methods were applied such as sequence typing, determination of virulence encoding genes and capsule locus.

According to the authors, results are convergent for all the species regardless the ANI threshold used expect for *Enterobacter hormaechei* genomes. One threshold defined one species divided into five subspecies, the other one three species. The authors argued that the approach species ANI > 95% and subspecies >98% should be preferred to the other one for two reasons. First, results of typing methods and phylogenetic approach suited best to the five subspecies classification. Then two strains were associated to two different species using the ANI > 96% for species definition which is not acceptable.

This article is interesting and well written. There are some comments/limitation which need to be corrected.

Major comments:

Lines 50 – 53, a very important taxonomic method among *Enterobacter cloacae* strains is based on the sequence of the gene *hsp60* described by Hoffman *et al* in 2003: Hoffmann, H., & Roggenkamp, A. (2003). Population genetics of the nomenclature *Enterobacter cloacae*. Applied and environmental microbiology, 69(9), 5306-5318. This approach needs to be present in this section.

Lines 53-54: “While MS has been commonly used in clinical laboratories, it has been 54 proven inaccurate for ECC classification”. Please add a citation to this affirmation.

Line 69: I don’t find any information about the Granger *et al* publication in the reference section.

Line 87: Is it possible to have a few clinical data about the demography of the patients? Thank you.

Line 76 – 77: “Zong *et al.* reclassified ECC species of ECC based on the threshold of ANI > 96% and 77 dDDH > 70.0% (36).” What were their conclusion? How many species in the complex? What were the consequences for *E. hormaechei* classification?

Line 88: Strains were isolated from 2010 to 2022. How were they conserved? What was the conservation medium? What temperature? What were the MALDI-TOF results for these strains?

Lines 92 – 98: **Sequencing, assembly, and annotation of bacterial genomes:** The absence of *hsp60* Hoffman cluster assignation is a limitation as this classification is widespread among *Enterobacter cloacae* complex studies and publications.

Line 96 – 97: “*Virulence genes were identified using VFDB databases (27)*” What blast parameters thresholds did you used for identity and coverage to consider a gene as present in the genome?

Line 100: “*the type strains of Enterobacter species and subspecies were collected from NCBI Refseq Database (24)*”. Could you please provide a list of the reference genomes you used as the reference for species and subspecies definition?

Line 103: The whole-genome phylogenetic analysis was performed kSNP3. Did you used a reference genome to annotate your SNP? If so, what was the reference chosen? What was the kmer size used? Did you run Kchooser first?

Line 107: BioProject accession PRJNA1046721. Data are not available at the moment of the review, could you please release at least the bioproject and the biosamples? Thank you.

Figure 1: Why some Sequence-type are missing? You need to declare all the new MLST into PubMLST. The figure cannot be acceptable with such missing data. If I understand correctly, Only the *E. hormaechei* are represented in this figure. This should be precise in the figure tittle.

Lines 126 – 133: **Phylogenetic analysis and sequence types (STs).** Is it possible to inform the genomic distance based on SNP between the five clades and within each clades using the results of kSNP3 ? You highlighted some outbreaks. What were the genomic distances between the outbreak strains?

Lines 135 – 148: **Virulence genes analysis.** The association between *iroBCDEN* genes clusters and *E. hormaechei* subsp. *steigerwaltii* is very interesting. Did you look at the outcome of the patients or the clinical presentation of the bacteraemia? You discussed the association with *E. bugandensis* and neonate’s mortality lines 200 – 201. This is why is it important to provide some clinical data associated with you strains.

Lines 163 – 169: **ANI**, interesting examples of taxonomy confusion. Did this problem occurred for more than two strains? Table I is informative.

Lines 172 – 175: “Accurate subspecies and species identification of important pathogens is pivotal for effective clinical decision-making in clinical practice (6, 7). With the tools of species classification continually improving (33). and a significant reduction in costs, ANI emerged as a widely utilized and recommended method for determining species boundaries.” These statements should be moderate. In everyday practical practice, all the strains are not sequenced. As the consequence “effective clinical decision” are not based on ANI results even if the cost of sequencing decreased significantly.

Line 185 – 189: “In contrast to the alignment-based approaches, like ANI, the k-mer-based kSNP approach exhibited a high contrast enhancement effect in analyzing highly similar sequences and identifying small differences between highly similar sequences, thereby providing a valuable perspective for constructing evolutionary trees with higher accuracy (11).” I did not see any comparison of clustering approaches in your study i.e. trees or distance matrix obtained by kSNP approach versus ANI approach. As far as I’m concerned you did not provide results to discuss the ability of the two methods to cluster genomic sequences.

Minor comment:

Line 46: The order of the bibliography is not appropriate. The number of the first article cited is 28

Line 51: please put a capital on the g of **G**ram staining

Line 63 and in diverse part of the manuscript: the expression “*et al*” needs to be written in italic.

Line 117 – 118: The most common species in clinical samples **was** *E. hormaechei* (Fig. 1)

Line 145: (update to September 2023)  September

Lines 150 – 161: Capsule typing. Maybe this paragraph should be shortened.

Dear editor,

We have revised the article (*Genome Sequence-Based Species Classification of Enterobacter cloacae complex: a Study among Clinical Isolates [Manuscript ID: Spectrum04312-23]*) according to the major concerns of the reviewers. Detailed descriptions of the changes being made in our current manuscript are given in the point-by-point account below.

We hope that our thoroughly revised manuscript is now acceptable for publication in the *Microbiology Spectrum*.

Yours sincerely,

Dr. Jiyong Yang

Reviewer 1:	
Reviewers' Comments	Reply to Comments
The manuscript entitled "Genome Sequence-Based Species Classification of Enterobacter cloacae complex: a Study among Clinical Isolates" reports a study, which is genome sequence-driven identification of 200+ clinical strains within the ECC. 1) However, the ANI values were used as the only deciding parameter for species and subspecies level identification of the bacterial strains.2) The unavailability of sequence accession numbers makes it more difficult to convince with the overall conclusions of the study.3) The study included ECC strains collected from the year 2010 to 2022, which is a big positive, but no information on the related metadata, like the origin of strains is missing. The initial identification was done by MALDI-TOF MS but the results were not discussed at all.	1) We thank the reviewer for pointing out this issue. Digital DNA: DNA hybridization (dDDH) was the major criterion used to justify ANI implementations and thresholds for species delineation, in accordance with the recommendations by the ad hoc committee for the re-evaluation of the species definition in bacteriology. For the ECC strains in our study, we also used dDDH for species classification, and the results were consistent with the results of species and subspecies classification using ANI value. We indeed should present this result in the study. We have revised it in METHOD section, Line134-135, in RESULT section, Line 160-162 and in DISCUSSION section, Line 264-269. 2) We have revised the release date and the sequence accession number is now available (BioProject accession: PRJNA1046721). 3) The isolates were stored in 30% glycerol at -80°C prior to further analysis. The duplicate samples were removed after the strain was resuscitated. 256 bloodstream isolated ECC was distributed at all ages, and the largest number of strains were middle-aged (41 to 65 years old) and elderly (>66 years old), with 116 strains (45.31%) and 94 strains (38.67%) respectively. The results on the use the MALDI-TOF MS method for fast identification within the ECC have demonstrated that it is inadequate for distinguishing E. asburiae, E. hormaechei, E. kobei and E. ludwigii from E. cloacae. We have revised it in Line 101-108.
1) L60-62: There are no two different ANI thresholds, it is just one range from 95-96%. As it has been found that	The statements have been corrected. We have revised it in the Line 67-70.

different methods and tools of ANI calculations can lead to variations, this range of ANI values is considered a threshold.	
2) Line 62: Several recent studies have suggested that the subspecies delineations are not accurate as there is no evidence of genetic isolation between them. One can take the example of Klebsiella pneumoniae, where three of its subspecies are now considered just Klebsiella pneumoniae. In the case of ECC, there is plenty of evidence that the five subspecies of E. hormaechei are divided into three species of Enterobacter, please refer to the latest research in this area.	We thank the reviewer for pointing out this issue. As we presented in INTRODUCTION section, Line 82-89, we also noticed that there is literature suggesting that the five subspecies of E. hormaechei are divided into three species of Enterobacter based on ANI > 96% and isDDH > 70%. However, in the process of molecular characterization of 256 clinical strains, our clinical data suggest that there is the association between iroBCDEN genes clusters and E. hormaechei subsp. steigerwaltii. For clinical strains, the classification of subspecies may be reasonable. This is where the confusion and the question comes in, which leads us to a deeper analysis.
3) Line 69: Granger et al. is missing from the list of references.	It has been modified.
4) Line 72-73: Add reference here	It has been added.
5) Line 73: The species "E. xiangfangensis" and "E. hoffmanii" are not yet validated, therefore always express them under "".	It has been modified.
6) L76-77: Both ANI and dDDH values indicated the species delineation, which is further substantiated by the recent core-gene phylogeny of ECC (see the recent publications on this).	We thank the reviewer for pointing out this issue. Digital DNA: DNA hybridization (dDDH) was the major criterion used to justify ANI implementations and thresholds for species delineation. We have revised it in the Line 82-84 and added the citation.
7) L78-79: Authors are wrongly interpreting the ANI range, I would like to mention once again this is a range not two different thresholds. This range helps when the different ANI values result from different ANI calculation tools, which follow slightly different methods of ANI calculations. The whole Material and Methods sections need more details, it is very brief in the present form.	The statements have been corrected. We have revised it in the Line 67-70, Line 82-84 and Line 91-92. The whole Material and Methods sections need more details, we have added it.
8) L94-95: There is a need to explain what are quality criteria used for taking the sequencing data further for assembly.	We have revised it in the Line 112-115.
9) L100-101: I think more information is required on how the dataset has been generated for type strains because in some instances NCBI RefSeq database can also have taxonomic errors. It is advised to define the whole process of dataset creation, which was further used for ANI calculations and phylogenetic analysis. I do not understand why the authors ignored the dDDH calculations, as it is still very relevant.	The following type strains were used for the ANI analysis: E. asburiae ATCC 35953 (CP011863), E. hormaechei ATCC 49162 (QZCT00000000), E. hormaechei subsp. steigerwaltii DSM16691 (AEXB00000000), E. hormaechei subsp. oharae DSM16687 (JCKW00000000), E. hormaechei subsp. hormaechei ATCC 49162 (QZCT00000000), E. hormaechei subsp. hoffmanii DSM 14563 (CP001918), E. hormaechei subsp. xiangfangensis LMG27195 (AP019007), E. kobei UCI 24 (POVL00000000), E. cloacae ATCC 13047 (MKEQ00000000), E. bugandensis EB-247 (QZCS00000000), E. cancerogenus JY65 (SJON00000000), E. chengduensis WCHEC1-C4 (CP017184), E. chuandaensis 090028 (CP017279), E. ludwigii EN-119 (LXES00000000), E. quasihormaechei WCHEQ120003 (SJO00000000), E.

	roggenkampii DSM 16690 (FYBI00000000), E. huaxiensis 090008 (CP017186), E. mori LMG 25706 (CP017179), E. oligotrophica CCA6 (MKEQ00000000), E. sichuanensis WCHECL1597 (CP017183), E. soli ATCC BAA-2102(CP017180), E. wuhouensis WCHEs120002 (CP011863). The dDDH among 256 clinical isolates were determined using the web-service https://tygs.dsmz.de/ (23). We have revised it in the Line 123-135.
10) L135-137: No information is provided on the sequence similarity cutoff used for virulence gene assignments.	Virulence genes were identified using VFDB databases (19), the coverage $\geq 75\%$ and identity $\geq 50\%$. We have added it in the Line 117.
11) L164-169: Authors are making an overstatement here, such situations are likely to happen, as there will be always a few strains that are almost on the border of a species, and in such cases the high closest similarity with enough dissimilarity with the second closest need to be considered before making conclusions. Secondly, there is a need to look into other parameters, like dDDH and phylogenetic position in core-gene trees or species trees.	Thanks for the reviewer's suggestions and we agree with your statements. We also mentioned this in the DISCUSSION section, Line 260, but it is still not comprehensive enough. We added our statements in Line 274-277.
Reviewer 2:	
Major points	Reply to Comments
1) Lines 50 – 53, a very important taxonomic method among Enterobacter cloacae strains is based on the sequence of the gene hsp60 described by Hoffman et al in 2003: Hoffmann, H., & Roggenkamp, A. (2003). Population genetics of the nomenspecies Enterobacter cloacae . Applied and environmental microbiology, 69(9), 5306-5318. This approach needs to be present in this section.	We thank the reviewer for pointing out this issue. We have revised it in the Line 62.
2) Lines 53-54: "While MS has been commonly used in clinical laboratories, it has been 54 proven inaccurate for ECC classification". Please add a citation to this affirmation.	We have added it in the Line 63.
3) Line 69: I don't find any information about the Granger et al publication in the reference section.	We have revised it in the Line 79.
4) Line 76 – 77: "Zong et al. reclassified ECC species of ECC based on the threshold of ANI > 96% and 77 dDDH > 70.0% (36)." What were their conclusion? How many species in the complex? What were the consequences for E. hormaechei classification?	We thank the reviewer for pointing out this issue. Zong et al. found that all Enterobacter subspecies assignments were incorrect. In this study, they updated the taxonomy of Enterobacter from species (n=19, including 6 subspecies) as of April 2020 to species (n=22). In other words, E. hormaechei were classified into three species (E. xiangfangensis , E. hoffmanii , and E. hormaechei). We have added it in the Line 87-89.
5) Line 87: Is it possible to have a few clinical data about the demography of the patients? Thank you.	Bloodstream isolated ECC was distributed at all ages, and the largest number of strains were middle-aged (41 to 65 years old) and elderly (>66 years old), with 116 strains (45.31%) and 94 strains (38.67%) respectively. We have added it in the Line 106-108.
6) Line 88: Strains were isolated from 2010 to 2022. How were	The isolates were stored in 30% glycerol at -80°C until further

they conserved? What was the conservation medium? What temperature? What were the MALDI-TOF results for these strains?	analysis. The isolates were identified as E. asburiae, E. hormaechei, E. kobei, E. ludwigii or E. cloacae because MALDI-TOF MS method is inadequate for distinguishing ECC. We have added it in the Line 101-102 and Line 103-105.
7) Lines 92 – 98: Sequencing, assembly, and annotation of bacterial genomes: The absence of hsp60 Hoffman cluster assignation is a limitation as this classification is widespread among Enterobacter cloacae complex studies and publications.	Thank you for your advice. There is no doubt that hsp60 Hoffman cluster assignation is very useful and convenient. It can be divided into ECC clusters by pcr method. In this paper, we conducted whole genome sequencing of strains, focusing on species classification by using the whole genome of clinical strains. Therefore, we did not include the hsp60 Hoffman cluster assignation in this paper.
8) Line 96 – 97: “Virulence genes were identified using VFDB databases (27)” What blast parameters thresholds did you used for identity and coverage to consider a gene as present in the genome?	We have added it in the Line 117.
9) Line 100: “the type strains of Enterobacter species and subspecies were collected from NCBI Refseq Database (24)”. Could you please provide a list of the reference genomes you used as the reference for species and subspecies definition?	We have added it in the Line 123-134.
10) Line 103: The whole-genome phylogenetic analysis was performed kSNP3. Did you used a reference genome to annotate your SNP? If so, what was the reference chosen? What was the kmer size used? Did you run Kchooser first?	The collected strains have been identified as ECC by MS, and the sequence similarity between strains is very high, so we did not use a reference genome. We ran the Kchooser to calculate the appropriate k-mer size, which is 19. We have added it in the Line 138.
11) Line 107: BioProject accession PRJNA1046721. Data are not available at the moment of the review, could you please release at least the bioproject and the biosamples? Thank you. Figure 1: Why some Sequence-type are missing? You need to declare all the new MLST into PubMLST. The figure cannot be acceptable with such missing data. If I understand correctly, Only the E. hormaechei are represented in this figure. This should be precise in the figure title.	We thank the reviewer for pointing out this issue. We have revised the release date and the sequence accession number is now available. All new MLST have been uploaded to PubMLST. Data submitted the website will go in to a queue for handling by a curator. New STs identified in this study are indicated in the figure using novel. The missing data in the figure has been added and the figure title has been modified.
12) Lines 126 – 133: Phylogenetic analysis and sequence types (STs). Is it possible to inform the genomic distance based on SNP between the five clades and within each clades using the results of kSNP3? You highlighted some outbreaks. What were the genomic distances between the outbreak strains?	The genomic distance has showed in the figure as 3 decimals kept. The outbreaks genomic distance is smaller than 0.0003. We have added it in the Line 168-170.
13) Lines 135 – 148: Virulence genes analysis. The association between iroBCDEN genes clusters and E. hormaechei subsp. steigerwaltii is very interesting. Did you look at the outcome of the patients or the clinical presentation of the bacteraemia? You discussed the association with E. bugandensis and neonate’s mortality lines 200 – 201. This is why is it important to provide some clinical data associated with you strains.	We looked at the clinical outcomes of patients with bacteremia in different subspecies of Enterobacter and found no significant differences in disease severity and mortality, regardless of whether this iroBCDEN virulence gene cluster was present. Our findings may have some implications, but there are many factors that influence the clinical presentation of patients with bacteremia. Thank you for your suggestion, which is of great clinical value and significance, and we will also pay attention to this aspect in the future. We have added it in the Line 188-190.
14) Lines 163 – 169: ANI, interesting examples of taxonomy confusion. Did this problem occurred for more than two	In our study, the problem of taxonomy confusion due to different ANI thresholds only occurred in these two strains of E. hormaechei. In other

strains? Table I is informative.	species of ECC, the ANI values varied greatly, and the genetic isolation between species was obvious.
15) Lines 172 – 175: “Accurate subspecies and species identification of important pathogens is pivotal for effective clinical decision-making in clinical practice (6, 7). With the tools of species classification continually improving (33). and a significant reduction in costs, ANI emerged as a widely utilized and recommended method for determining species boundaries.” These statements should be moderate. In everyday practical practice, all the strains are not sequenced. As the consequence “effective clinical decision” are not based on ANI results even if the cost of sequencing decreased significantly.	The statements have been corrected, these statements should be moderate. We have revised it in the Line 213-214.
16) Line 185 – 189: “In contrast to the alignment-based approaches, like ANI, the k-mer-based kSNP approach exhibited a high contrast enhancement effect in analyzing highly similar sequences and identifying small differences between highly similar sequences, thereby providing a valuable perspective for constructing evolutionary trees with higher accuracy (11).” I did not see any comparison of clustering approaches in your study i.e. trees or distance matrix obtained by kSNP approach versus ANI approach. As far as I’m concerned you did not provide results to discuss the ability of the two methods to cluster genomic sequences.	We thank the reviewer for pointing out this issue. There was some inappropriateness in our presentation. We did ANI clustering analysis and the results of 162 E.hormaechei clinical isolates were consistent with kSNP approach. Tree matrix obtained by ANI approach are shown in the figure below. We just wanted to provide a multi-dimensional approach to species classification in addition to ANI clustering analysis, so we did a k-mer-based kSNP phylogenetic analysis. We have revised our statements in the Line 225, Line 229-231. Minor points	Reply to Comments
1) Line 46: The order of the bibliography is not appropriate. The number of the first article cited is 28 Line 51: please put a capital on the g of Gram staining Line 63 and in diverse part of the manuscript: the expression “et al” needs to be written in italic. 2) Line 117 – 118: The most common species in clinical samples was E. hormaechei (Fig. 1) Line 145: (update to September 2023)  September 3) Lines 150 – 161: Capsule typing. Maybe this paragraph should be shortened.	It has been modified.

Re: Spectrum04312-23R1 (Genome Sequence-Based Species Classification of *Enterobacter cloacae* complex: a Study among Clinical Isolates)

Dear Dr. Jiyong Yang:

Thank you for the privilege of reviewing your work. Below you will find my comments, instructions from the Spectrum editorial office, and the reviewer comments.

Revision Guidelines

Sincerely,
Sadjia Bekal
Editor
Microbiology Spectrum

Reviewer #1 (Comments for the Author):

I believe the authors still do not agree that the recent classifications within ECC were not solely based on ANI values; instead, they are based on phylogenetic congruence and various genomic features, including ANI. However, if there is something wrong with one or the other ECC taxonomy, it should not influence this study. As far as I understood, this study does not aim to define the species and subspecies within *E. hormaechei*, but rather to report on the identification of ECC isolates. If the authors

consider subspecies-based taxonomy to be accurate, then they should discuss their results accordingly. Defining *E. hormaechei* falls outside the scope of this study, and the authors have not conducted any tests to confirm which taxonomy is better or more suitable. If authors, aim to define this species, or the whole ECC, they need to generate scientific arguments based on strong experimental data. Furthermore, linking *iroBCDEN* genes clusters and *E. hormaechei* subsp. *steigerwaltii*, was not done appropriately like this linkage was noticed only for the genomes sequenced during this study, and authors do not include genomes of other *E. hormaechei* subsp. *steigerwaltii* strains. Secondly, the criteria for the identification of virulence factors (genes) was quite relaxed (the coverage {greater than or equal to}75% and identity {greater than or equal to}50%). Additionally, authors mentioned that kSNP (k-mers-based) phylogenetic tree is better, and I also agree with it, however, the choice depends on your specific research questions. If authors, just want to represent the genetic diversity within the closely related strains. In this case, kSNP might capture sufficient variation, but for species-level variations, a core-gene phylogeny might be more informative. Overall, I suggest that the authors need to change the focus of this study and establish a direct link between the methods used and the results obtained, else authors should use more robust methods and obtain new results, which provide strong evidence for their conclusions.

Reviewer #2 (Comments for the Author):

Thank you very much for your article. Its improved a lot from the first version. I still have three major comments, two from the results and the last one from the discussion part.

Review of: Genome Sequence-Based Species Classification of *Enterobacter cloacae* complex: a Study among Clinical Isolates

Qiu and colleagues compared two taxonomic approaches for *Enterobacter cloacae* complex genomes. These two were based on Average Nucleotide Identity (ANI) but the thresholds applied for the species definition could be: ANI > 95% and subspecies >98% or ANI > 96% for species definition and no data for subspecies.

To test the two thresholds, they used a collection of 256 clinical strains isolated from blood cultures between 2010 and 2022. A global phylogenetic approach was performed using the kSNP. For each genome, several typing methods were applied such as sequence typing, determination of virulence encoding genes and capsule locus.

According to the authors, results are convergent for all the species regardless the ANI thresholds used expect for *Enterobacter hormaechei* genomes. One threshold defined one species divided into five subspecies, the other one three species. The authors argued that the approach species ANI > 95% and subspecies >98% should be preferred to the other one for two reasons. First, results of typing methods and phylogenetic approach suited best to the five subspecies classification. Secondly, two strains were associated to two different species using the ANI > 96% for species definition which is not acceptable.

This article is interesting and well written and improved a lot from the first version.

There are some comments/limitations which need to be corrected:

The important part needs to be rewrite, this part should explain why the article is important. In its present form, the text looks like a small abstract of the article.

Major comment:

In the figure 1: Previously *unknown* sequence types are now labelled “*novel*”; this is not enough. News sequence types must be declared into PubMLST database to obtain proper attribution numbers. This is an important limitation for readers, they can’t easily compare their genomes with those from the publication without proper ST declaration.

Lines 168 – 170: “*The genomic distance between the five clades and within each clade has showed in the figure as 3 decimals kept (Fig. 1). The outbreaks genomic distance is smaller than 0.0003.*” I’m not familiar with this distance metric and didn’t see any information about it in the methods part. Could you please explain how it was calculated? SNP counts were unavailable from the kSNP3 output files?

Line 245 – 246 “*This underscores the clinical significance of distinguishing strains with and without specific virulence gene clusters, such as *iroN* gene clusters, in clinical pathogenicity analyses.*” But according to the results section, you didn’t find any difference in outcome for your patient according to the ECC species i.e the presence of *iroN* genes clusters. This should be discussed.

Minor comments:

Line 63, genes need to be written in italic: *hsp60*.

Lines 76 – 77 “Chavda et al. employed single nucleotide polymorphisms 76 (SNPs) from whole genome alignments to establish groups within ECC (13).”. What were their conclusions? How many groups did they obtained?

Lines 83 – 84 “Recently, researchers proposed a novel perspective (Species: ANI > 96% and dDDH > 70%), classifying *E. hormaechei* into three species: *E. hormaechei*, “*E. xiangfangensis*”, and “*E. hoffmanii*” (15, 16).” Could you please explain the scientific reasons why these authors changed the ANI threshold? Why didn’t they accept the classification of Sutton *et al*?

Lines 104 – 105 “*because MALDI-TOF MS method is inadequate for distinguishing ECC*”. Seems to me unnecessary.

Line 112: The SPAdes version is unknown.

Line 120: ANI and digital DNA:DNA hybridization (dDDH) analysis. It is possible to put the data into a data frame?

Line 174- 176: “We annotated siderophores-related genes, including entABESfepABCDG (enterobactin), iutAiuCABCD (aerobactin), and *iroBCDEN* (salmochelin), across the 162 clinical isolates”. This should be in the methods part. The non-siderophores-related genes are not mentioned in the text which is surprising. Does it mean that the genomes only contained siderophores-related genes as virulence genes?

Line 231: *E. hormaechei* needs to be in italic.

Dear editor,

We have revised the article (*Genome Sequence-Based Species Classification of Enterobacter cloacae complex: a Study among Clinical Isolates [Manuscript ID: Spectrum04312-23R1]*) according to the major concerns of the reviewers. Detailed descriptions of the changes being made in our current manuscript are given in the point-by-point account below.

We hope that our thoroughly revised manuscript is now acceptable for publication in the *Microbiology Spectrum*.

Yours sincerely,

Dr. Jiyong Yang

Reviewer 1:	
Reviewers' Comments	Reply to Comments
I believe the authors still do not agree that the recent classifications within ECC were not solely based on ANI values; instead, they are based on phylogenetic congruence and various genomic features, including ANI. However, if there is something wrong with one or the other ECC taxonomy, it should not influence this study. As far as I understood, this study does not aim to define the species and subspecies within E. hormaechei, but rather to report on the identification of ECC isolates. If the authors consider subspecies-based taxonomy to be accurate, then they should discuss their results accordingly. Defining E. hormaechei falls outside the scope of this study, and the authors have not conducted any tests to confirm which taxonomy is better or more suitable. If authors, aim to define this species, or the	Thank you for your careful review and valuable suggestions on this research work. As you pointed out, our study is aiming to discuss that the taxonomy based on subspecies is more suitable and valuable for the clinical isolates, but not to evaluate the accuracy of the classification criteria to identify species/subspecies. However, some expressions in the manuscript may be ambiguous and prone to misunderstanding. We have modified the manuscript thoroughly. 1. In our study, we found that taxonomy based on subspecies is more suitable for clinical isolates. It can differentiate strains with multiple shared characteristics, including virulence genes and capsule types. Our results indicate that the identification result of clinical strains based on subspecies is more suitable for subsequent analysis such as molecular features, pathogenicity and virulence characteristics. The species-level identification may obscure the differences in pathogenicity and virulence characteristics among different

whole ECC, they need to generate scientific arguments based on strong experimental data.

Furthermore, linking *iroBCDEN* genes clusters and *E. hormaechei* subsp. *steigerwaltii*, was not done appropriately like this linkage was noticed only for the genomes sequenced during this study, and authors do not include genomes of other *E. hormaechei* subsp. *steigerwaltii* strains. Secondly, the criteria for the identification of virulence factors (genes) was quite relaxed (the coverage {greater than or equal to}75% and identity {greater than or equal to}50%).

Additionally, authors mentioned that kSNP (k-mers-based) phylogenetic tree is better, and I also agree with it, however, the choice depends on your specific research questions. If authors, just want to represent the genetic diversity within the closely related strains. In this case, kSNP might capture sufficient variation, but for species-level variations, a core-gene phylogeny might be more informative.

Overall, I suggest that the authors need to change the focus of this study and establish a direct link between the methods used and the results obtained, else authors should use more robust methods and obtain new results, which provide strong evidence for their conclusions.

subspecies. All current classification criteria are undisputable and have their application scenarios. The aim of this study is not to establish classification criteria, but rather to evaluate the practicality of various taxonomy in the current application scenario (clinical strains isolated from bloodstream). Utilizing established molecular identification techniques including ANI, dDDH and phylogenetic analysis, we sought to find the most appropriate classification criteria for our clinical strains. Corresponding content has been modified in **ABSTRACT section, Line 28-40, INTRODUCTION section, Line 69-72, Line 99-104, DISCUSSION section, Line 210-212 and CONCLUSION section, Line 281-282.**

2. We fully agree that the classifications within ECC were based on many dimensions including ANI, dDDH and phylogenic analysis. The point of this manuscript is to discuss the application of the taxonomy in clinical scenario but not to questioning the classification criterion itself. It has been modified in **ABSTRACT section, Line 24-28, in INTRODUCTION section, Line 66-69, and in DISCUSSION section, Line 209-210.**

3. In order to solidify the relationship between *iroBCDEN* genes and *E. hormaechei* subsp. *steigerwaltii*, we also analyzed 6221 genomes of *E. hormaechei* from the NCBI database (update to September 2023). The results revealed that *iroBCDEN* genes were only found in 2722 *E. hormaechei* subsp. *steigerwaltii* genomes but not in other four subspecies. The 3rd generation sequencing results indicated that the *iroBCDEN* located in the chromosome of the *E. hormaechei* subsp. *steigerwaltii*, which is not easily transferable as plasmid. **It has been shown in RESULT section, Line 177-181.**

Furthermore, we adjusted the thresholds for the coverage and identity of virulence genes to 80% and 90%, respectively. Our analysis

	revealed that the comparison results for the iroBCDEN genes cluster exhibit the coverage > 99.8% and the identity > 91%. It has been shown in METHODS section, Line 121-122. 4. We agree with your perspective that for variation at the species level, the core-gene phylogenetic tree could provide more information. kSNP is a software that simultaneously identifies SNPs based on both the core genome and the whole genome across a set of sequenced genomes. These tree files are then reported in the output files. In our study, we sought to characterize the species-level variation, thus we constructed phylogenetic trees using core SNPs. It has been modified in METHODS section, Line 131-132, in DISCUSSION section, Line 217-222.
Reviewer 2:	
Major points	Reply to Comments
The important part needs to be rewrite, this part should explain why the article is important. In its present form, the text looks like a small abstract of the article.	We have modified it in the Line 42-49 .
In the figure 1: Previously unknown sequence types are now labelled “novel”; this is not enough. News sequence types must be declared into PubMLST database to obtain proper attribution numbers. This is an important limitation for readers, they can’t easily compare their genomes with those from the publication without proper ST declaration.	New alleles were identified and assigned new allele numbers by comparison against the pubMLST database. All new allele sequences are deposited at the public databases for molecular typing and microbial genome diversity (pubMLST). The allelic profiles of strains are marked in red in Fig 1 .
Lines 168 – 170: “The genomic distance between the five clades and within each clade has showed in the figure as 3 decimals kept (Fig. 1). The outbreaks genomic distance is smaller than 0.0003.” I’m not familiar with this distance metric and didn’t see any information about it in the methods part. Could you	As you pointed out, the genomic genetic distance between species calculated based on the SNP counts and the nucleotide substitution model. The kSNP3 output files include SNP count files for branches. The genomic distance based on SNP between the five clades and within each clade are illustrated in the figure (Fig. 1). The identification of outbreak strains should be based on clinical

please explain how it was calculated? SNP counts were unavailable from the kSNP3 output files?	epidemiological analysis, which includes considerations of isolate time, location, and phylogenetic distance. We previously assumed that strains with a very close genetic distance on the evolutionary tree represented an outbreak, which may cause controversy. Therefore, we have removed the relevant content from the manuscript to avoid potential ambiguity.. We have revised it in the Line 161-162.
Line 245 – 246 “This underscores the clinical significance of distinguishing strains with and without specific virulence gene clusters, such as iroN gene clusters, in clinical pathogenicity analyses.” But according to the results section, you didn’t find any difference in outcome for your patient according to the ECC species i.e the presence of iroN genes clusters. This should be discussed.	There were no significant differences in disease severity and mortality among patients with bacteremia of different subspecies of E. hormaechei, regardless of the presence or absence of the iroBCDEN gene cluster. The pathogenicity of the strains is influenced by many factors and regulatory mechanisms. The presence of salmochelin in our isolates may not affect the outcomes of patient. Siderophore can promote the survival and reproduction of bacteria in the host. We will further investigate the impact of this gene cluster on the survival and transmission of this strains. It has been modified in Line 239-243.
Minor points	Reply to Comments
1)Line 63, genes need to be written in italic: hsp60. 2)Lines 76 – 77 “Chavda et al. employed single nucleotide polymorphisms (SNPs) from whole genome alignments to establish groups within ECC (13).”. What were their conclusions? How many groups did they obtained? Lines 83 – 84 “Recently, researchers proposed a novel perspective (Species: ANI > 96% and dDDH > 70%), classifying E. hormaechei into three species: E. hormaechei, “E. xiangfangensis”, and “E. hoffmanii” (15, 16).” Could you please explain the	It has been modified. Chavda et al. employed ANI and single nucleotide polymorphisms (SNPs) from whole genome alignments to establish 18 groups (groups A to R) within ECC (13). They found the mean ANI values between 18 ECC groups were always $\leq 95\%$, except among E. hormaechei subspecies groups A to E, G, and H. It has been modified in Line 76-79. The reason for changing the ANI threshold is based on previous research. Ciufu et al. found that ANI > 96% can discriminate the most strains (prokaryotic genomes in the NCBI) into distinct species, and it is observed that a lower ANI value correlates with an increased likelihood of ambiguous classification outcomes. (CIUFO S, KANNAN S,

scientific reasons why these authors changed the ANI threshold? Why didn't they accept the classification of Sutton et al?	SHARMA S, et al. Using average nucleotide identity to improve taxonomic assignments in prokaryotic genomes at the NCBI [J]. Int J Syst Evol Microbiol) Zong et al. used cutoff of ANI $\geq 96\%$ and isDDH $\geq 70.0\%$ for the type strain of Enterobacter (E. hormaechei subsp. oharae DSM 16687, E. hormaechei subsp. steigerwaltii DSM 16691, E. xiangfangensis LMG 27195). The ANI of three type strains were $\geq 96.62\%$, and the isDDH were $\geq 75.8\%$. Consequently, they classified the three type strains as one species. It has been modified in Line 89-95.
Lines 104 – 105 “because MALDI-TOF MS method is inadequate for distinguishing ECC”. Seems to me unnecessary.	It has been modified.
Line 112: The SPAdes version is unknown.	It has been modified in Line 118.
Line 120: ANI and digital DNA:DNA hybridization (dDDH) analysis. It is possible to put the data into a data frame?	The data has been put into Supplemental Material Table S1.
Line 174- 176: “We annotated siderophores-related genes, including entABESfepABCDG (enterobactin), iutAiuCABCD (aerobactin), and iroBCDEN (salmochelin), across the 162 clinical isolates”. This should be in the methods part. The non-siderophores-related genes are not mentioned in the text which is surprising. Does it mean that the genomes only contained siderophores-related genes as virulence genes?	Virulence genes were identified using VFDB databases, it has been shown in the methods part. We found virulence genes (encoding fimbriae, curli fibers, capsules, and flagella) were widely distributed in 162 strains with no significant differences. Siderophore-related genes (entABESfepABCDG (enterobactin) and iutAiuCABCD (aerobactin)) were present in all 162 clinical isolates. It has been modified in Line 166-170.
Line 231: E. hormaechei needs to be in italic.	It has been modified.

Re: Spectrum04312-23R2 (Genome Sequence-Based Species Classification of Enterobacter cloacae complex: a Study among Clinical Isolates)

Dear Dr. Jiyong Yang:

Your manuscript has been accepted, and I am forwarding it to the ASM production staff for publication. Your paper will first be checked to make sure all elements meet the technical requirements. ASM staff will contact you if anything needs to be revised before copyediting and production can begin. Otherwise, you will be notified when your proofs are ready to be viewed.

Sincerely,
Sadjia Bekal
Editor
Microbiology Spectrum